# A Way out of the Odyssey: Analyzing and Combining Recent Insights for LSTMs

**Shayne Longpre**
Salesforce Research
Palo Alto, California
slongpre@cs.stanford.edu

**Sabeek Pradhan**
Stanford University
Palo Alto, California
sabeekp@cs.stanford.edu

**Caiming Xiong, Richard Socher**
Salesforce Research
Palo Alto, California
{cxiong,rsocher}@salesforce.com

## Abstract

LSTMs have become a basic building block for many deep NLP models. In recent years, many improvements and variations have been proposed for deep sequence models in general, and LSTMs in particular. We propose and analyze a series of augmentations and modifications to LSTM networks resulting in improved performance for text classification datasets. We observe compounding improvements on traditional LSTMs using Monte Carlo test-time model averaging, average pooling, and residual connections, along with four other suggested modifications. Our analysis provides a simple, reliable, and high quality baseline model.

## 1 Introduction

When exploring a new problem, having a simple yet competitive off-the-shelf baseline is fundamental to new research. For instance, Caruana et al. (2008) showed random forests to be a strong baseline for many high-dimensional supervised learning tasks. For computer vision, off-the-shelf convolutional neural networks (CNNs) have earned their reputation as a strong baseline (Sharif Razavian et al., 2014) and basic building block for more complex models like visual question answering (Xiong et al., 2016). For natural language processing (NLP) and other sequential modeling tasks, recurrent neural networks (RNNs), and in particular Long Short-Term Memory (LSTM) networks, with a linear projection layer at the end have begun to attain a similar status. However, the standard LSTM is in many ways lacking as a baseline. Zaremba (2015), Gal (2015), and others show that large improvements are possible using a forget bias, inverted dropout regularization or bidirectionality. We add three major additions with similar improvements to off-the-shelf LSTMs: Monte Carlo model averaging, embed average pooling, and residual connections. We analyze these and other more common improvements.

## 2 LSTM Network

LSTM networks are among the most commonly used models for tasks involving variable-length sequences of data, such as text classification. The basic LSTM layer consists of six equations:

$$i_t = \tanh\left(W_i x_t + R_i h_{t-1} + b_i\right) \tag{1}$$

$$j_t = \sigma\left(W_j x_t + R_j h_{t-1} + b_j\right) \tag{2}$$

$$f_t = \sigma\left(W_f x_t + R_f h_{t-1} + b_f\right) \tag{3}$$

$$o_t = \tanh\left(W_o x_t + R_o h_{t-1} + b_o\right) \tag{4}$$

$$c_t = i_t \odot j_t + f_t \odot c_{t-1} \tag{5}$$

$$h_t = o_t \odot \tanh\left(c_t\right) \tag{6}$$

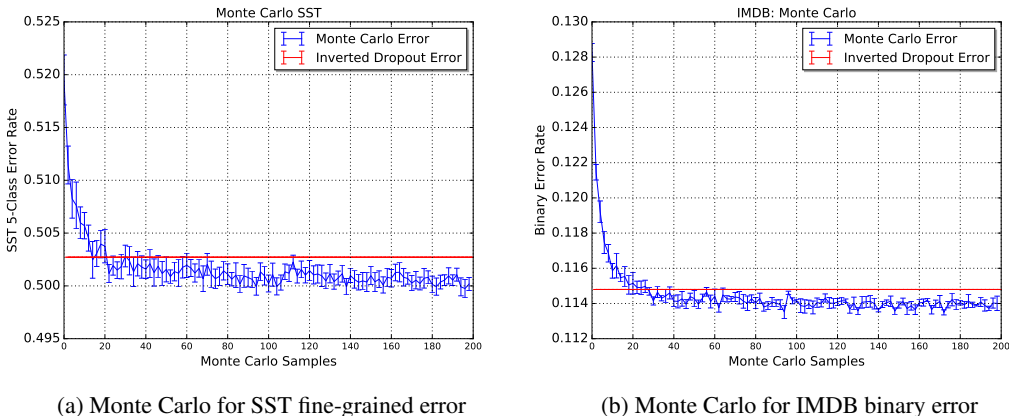

(a) Monte Carlo for SST fine-grained error (b) Monte Carlo for IMDB binary error

Figure 1: A comparison of the performance of Monte Carlo averaging, over sample size, to regular single-sample inverted dropout at test-time.

Where $\sigma$ is the sigmoid function, $\odot$ is element-wise multiplication, and $v_t$ is the value of variable $v$ at timestep $t$. Each layer receives $x_t$ from the layer that came before it and $h_{t-1}$ and $c_{t-1}$ from the previous timestep, and it outputs $h_t$ to the layer that comes after it and $h_t$ and $c_t$ to the next timestep. The $c$ and $h$ values jointly constitute the recurrent state of the LSTM that is passed from one timestep to the next. Since the $h$ value completely updates at each timestep while the $c$ value maintains part of its own value through multiplication by the forget gate $f$, $h$ and $c$ complement each other very well, with $h$ forming a "fast" state that can quickly adapt to new information and $c$ forming a "slow" state that allows information to be retained over longer periods of time (Zaremba, 2015). While various papers have tried to systematically experiment with the 6 core equations constituting an LSTM (Greff et al., 2015; Zaremba, 2015), in general the basic LSTM equations have proven extremely resilient and, if not optimal, at least a local maximum.

## 3 MONTE CARLO MODEL AVERAGING

It is common practice when applying dropout in neural networks to scale the weights up at train time (inverted dropout). This ensures that the expected magnitude of the inputs to any given layer are equivalent between train and test, allowing for an efficient computation of test-time predictions. However, for a model trained with dropout, test-time predictions generated without dropout merely approximate the ensemble of smaller models that dropout is meant to provide. A higher fidelity method requires that test-time dropout be conducted in a manner consistent with how the model was trained. To achieve this, we sample $k$ neural nets with dropout applied for each test example and average the predictions. With sufficiently large $k$ this Monte Carlo average should approach the true model average (Srivastava et al., 2014). We show in Figure 1 that this technique can yield more accurate predictions on test-time data than the standard practice. This is demonstrated over a number of datasets, suggesting its applicability to many types of sequential architectures. While running multiple Monte Carlo samples is more computationally expensive, the overall increase is minimal as the process is only run on test-time forward passes and is highly parallelizable. We show that higher performance can be achieved with relatively few Monte Carlo samples, and that this number of samples is similar across different NLP datasets and tasks.

We encountered one ambiguity of Monte Carlo model averaging that to our knowledge remains unaddressed in prior literature: there is relatively little exploration as to where and how the model averaging is most appropriately handled. We investigated averaging over the output of the final recurrent layer (just before the projection layer), over the output of the projection layer (the pre-softmax unnormalized logits), and the post-softmax normalized probabilities, which is the approach taken by Gal (2015) for language modeling. We saw no discernible difference in performance between averaging the pre-projection and post-projection outputs. Averaging over the post-softmax probabilities showed marginal improvements over these two methods, but interestingly only for bidirectional models. We also explored using majority voting among the sampled models. This

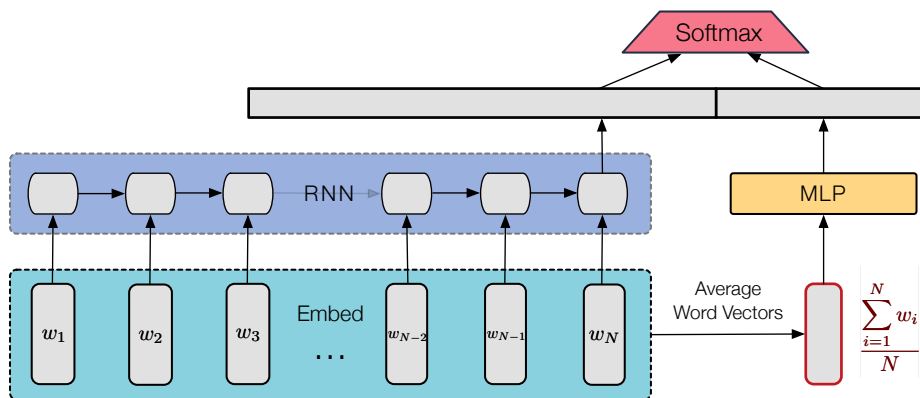

Figure 2: An illustration of the embed average pooling extension to a standard RNN model. The output of the multilayer perceptron is concatenated to the final hidden state output by the RNN.

involves tallying the maximum post-softmax probabilities and selecting the class that received the most votes. This method differs from averaging the post-softmax probabilities in the same way max-margin differs from maximum likelihood estimation (MLE), de-emphasizing the points well inside the decision boundary or the models that predicted a class with extremely high probability. With sufficiently large $k$, this voting method seemed to work best of the averaging methods we tried, and thus all of our displayed models use this technique. However, for classification problems with more classes, more Monte Carlo samples might be necessary to guarantee a meaningful plurality of class predictions. We conclude that the majority-vote Monte Carlo averaging method is preferable in the case where the ratio of Monte Carlo samples to number of classification labels is large ($k/output\_size$).

The Monte Carlo model averaging experiments, shown in Figure 1, were conducted as follows. We drew $k = 400$ separate test samples for each example, differentiated by their dropout masks. For each sample size $p$ (whose values, plotted on the x-axis, were in the range from 2 to 200 with step-size 2) we selected $p$ of our $k$ samples randomly without replacement and performed the relevant Monte Carlo averaging technique for that task, as discussed above. We do this $m = 20$ times for each point, to establish the mean and variance for that number of Monte Carlo iterations/samples $p$. The variance is used to visualize the 90% confidence interval in blue, while the red line denotes the test accuracy computed using the traditional approximation method (inverted dropout at train-time, and no dropout at test-time).

## 4 EMBED AVERAGE POOLING

Reliably retaining long-range information is a well documented weakness of LSTM networks (Karpathy et al., 2015). This is especially the case for very long sequences like the IMDB sentiment dataset (Maas et al., 2011), where deep sequential models fail to capture uni- and bi-gram occurrences over long sequences. This is likely why $n$-gram based models, such as a bi-gram NBSVM (Wang and Manning, 2012), outperform RNN models on such datasetes. It was shown by Iyyer et al. (2015) and others that for general NLP classification tasks, the use of a deep, unordered composition (or bag-of-words) of a sequence can yield strong results. Their solution, the deep averaging network (DAN), combines the observed effectiveness of depth, with the unreasonable effectiveness of unordered representations of long sequences.

We suspect that the primary advantage of DANs is their ability to keep track of information that would have otherwise been forgotten by a sequential model, such as information early in the sequence for a unidirectional RNN or information in the middle of the sequence for a bidirectional RNN. Our embed average pooling supplements the bidirectional RNN with the information from a DAN at a relatively negligible computational cost.

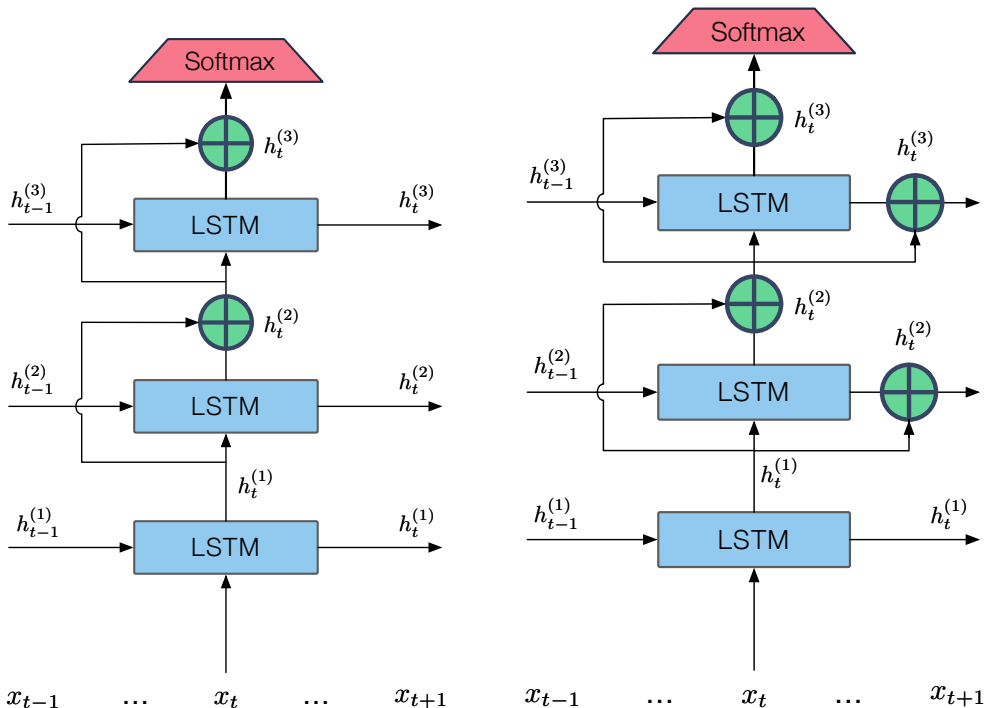

(a) Res-V1: An illustration of vertical residual connections

(b) Res-V2: An illustration of vertical and lateral residual connections

Figure 3: An illustration of vertical (ResV) and lateral residual (ResL) connections added to a 3-layer RNN. A model with only vertical residuals is denoted Res-V1, whereas a model with vertical and lateral residuals is denoted "Res-V2".

As shown in Figure 2, embed average pooling works by averaging the sequence of word vectors and passing this average through an MLP. The averaging is similar to an average pooling layer in a CNN (hence the name), but with the averaging being done temporally rather than spatially. The output of this MLP is concatenated to the final output of the RNN, and the combined vector is then passed into the projection and softmax layer. We apply the same dropout mask to the word vectors when passing them to the RNN as when averaging them, and we apply a different dropout mask on the output of the MLP. We experimented with applying the MLP before rather than after averaging the word vectors but found the latter to be most effective.

# 5 RESIDUAL CONNECTIONS

For feed-forward convolutional neural networks used in computer vision tasks, residual networks, or ResNets, have obtained state of the art results (He et al., 2015). Rather than having each layer learn a wholly new representation of the data, as is customary for neural networks, ResNets have each layer (or group of layers) learn a residual which is added to the layer's input and then passed on to the next layer. More formally, if the input to a layer (or group of layers) is $x$ and the output of that layer (or group of layers) is $F(x)$, then the input to the next layer (or group of layers) is $x + F(x)$, whereas it would be $F(x)$ in a conventional neural network. This architecture allows the training of far deeper models. He et al. (2015) trained convolutional neural networks as deep as 151 layers, compared to 16 layers used in VGGNets (Simonyan and Zisserman, 2014) or 22 layers used in GoogLeNet (Szegedy et al., 2015), and won the 2015 ImageNet Challenge. Since then, various papers have tried to build upon the ResNet paradigm (Huang et al., 2016; Szegedy et al., 2016), and various others have tried to create convincing theoretical reasons for ResNet's success (Liao and Poggio, 2016; Veit et al., 2016).

We explored many different ways to incorporate residual connections in an RNN. The two most successful ones, which we call Res-V1 and Res-V2 are depicted in Figure 6. Res-V1 incorporates only vertical residuals, while Res-V2 incorporates both vertical and lateral residuals. With vertical residual connections, the input to a layer is added to its output and then passed to the next layer, as is done in feed-forward ResNets. Thus, whereas the input to a layer is normally the $h_t$ from the previous layer, with vertical residuals the input becomes the $h_t + x_t$ from the previous layer. This maintains many of the attractive properties of ResNets (e.g. unimpeded gradient flow across layers, adding/averaging the contributions of each layer) and thus lends itself naturally to deeper networks. However, it can interact unpredictably with the LSTM architecture, as the "fast" state of the LSTM no longer reflects the network's full representation of the data at that point. To mitigate this unpredictability, Res-V2 also includes lateral residual connections. With lateral residual connections, the input to a layer is added to its output and then passed to the next timestep as the fast state of the LSTM. It is equivalent to replacing equation 6 with $h_t = o_t \odot \tanh(c_t) + x_t$. Thus, applying both vertical and lateral residuals ensures that the same value is passed both to the next layer as input and to the next timestep as the "fast" state.

In addition to these two, we explored various other, ultimately less successful, ways of adding residual connections to an LSTM, the primary one being horizontal residual connections. In this architecture, rather than adding the input from the previous layer to a layer's output, we added the fast state from the previous timestep. The hope was that adding residual connections across timesteps would allow information to flow more effectively across timesteps and thus improve the performance of RNNs that are deep across timesteps, much as ResNets do for networks that are deep across layers. Thus, we believed horizontal residual connections could solve the problem of LSTMs not learning long-term dependencies, the same problem we also hoped to mitigate with embed average pooling. Unfortunately, horizontal residuals failed, possibly because they blurred the distinction between the LSTM's "fast" state and "slow" state and thus prevented the LSTM from quickly adapting to new data. Alternate combinations of horizontal, vertical, and lateral residual connections were also experimented with but yielded poor results.

# 6 EXPERIMENTAL RESULTS

## 6.1 DATASETS

We chose two commonly used benchmark datasets for our experiments: the Stanford Sentiment Treebank (SST) (Socher et al., 2013) and the IMDB sentiment dataset (Maas et al., 2011). This allowed us to compare the performance of our models to existing work and review the flexibility of our proposed model extensions across fairly disparate types of classification datasets. SST contains relatively well curated, short sequence sentences, in contrast to IMDB's comparatively colloquial and lengthy sequences (some up to $2,000$ tokens). To further differentiate the classification tasks we chose to experiment with fine-grained, five-class sentiment on SST, while IMDB only offered binary labels. For IMDB, we randomly split the training set of $25,000$ examples into training and validation sets containing $22,500$ and $2,500$ examples respectively, as done in Maas et al. (2011).

## 6.2 METHODOLOGY

Our objective is to show a series of compounding extensions to the standard LSTM baseline that enhance accuracy. To ensure scientific reliability, the addition of each feature is the only change from the previous model (see Figures 4 and 5). The baseline model is a 2-layer stacked LSTM with hidden size 170 for SST and 120 for IMDB, as used in Tai et al. (2015). All models in this paper used publicly available 300 dimensional word vectors, pre-trained using Glove on 840 million tokens of Common Crawl Data (Pennington et al., 2014), and both the word vectors and the subsequent weight matrices were trained using Adam with a learning rate of $10^{-4}$.

The first set of basic feature additions were adding a forget bias and using dropout. Adding a bias of $1.0$ to the forget gate (i.e. adding $1.0$ to the inside of the sigmoid function in equation 3) improves results across NLP tasks, especially for learning long-range dependencies (Zaremba, 2015). Dropout (Srivastava et al., 2014) is a highly effective regularizer for deep models. For SST and IMDB we used grid search to select dropout probabilities of $0.5$ and $0.7$ respectively, applied to the input of each layer, including the projection/softmax layer. While forget bias appears to hurt performance in Figure

5, the combination of dropout and forget bias yielded better results in all cases than dropout without forget bias. Our last two basic optimizations were increasing the hidden sizes and then adding shared-weight bidirectionality to the RNN. The hidden sizes for SST and IMDB were increased to 800 and 360 respectively; we found significantly diminishing returns to performance from increases beyond this. We chose shared-weight bidirectionality to ensure the model size did not increase any further. Specifically, the forward and backward weights are shared, and the input to the projection/softmax layer is a concatenation of the forward and backward passes' final hidden states.

All of our subsequent proposed model extensions are described at length in their own sections. For both datasets, we used 60 Monte Carlo samples, and the embed average pooling MLP had one hidden layer and both a hidden dimension and an output dimension of 300 as the output dimension of the embed average pooling MLP. Note that although the MLP weights increased the size of their respective models, this increase is negligible (equivalent to increasing the hidden size for SST from 800 to 804 or the hidden size of IMDB from 360 to 369), and we found that such a size increase had no discernible effect on accuracy when done without the embed average pooling.

## 6.3 RESULTS

Since each of our proposed modifications operate independently, they are well suited to use in combination as well as in isolation. In Figures 4 and 5 we compound these features on top of the more traditional enhancements. Due to the expensiveness of bidirectional models, Figure 4 also shows these compounding features on SST with and without bidirectionality. The validation accuracy distributions show that each augmentation usually provides some small but noticeable improvement on the previous model, as measured by consistent improvements in mean and median accuracy.

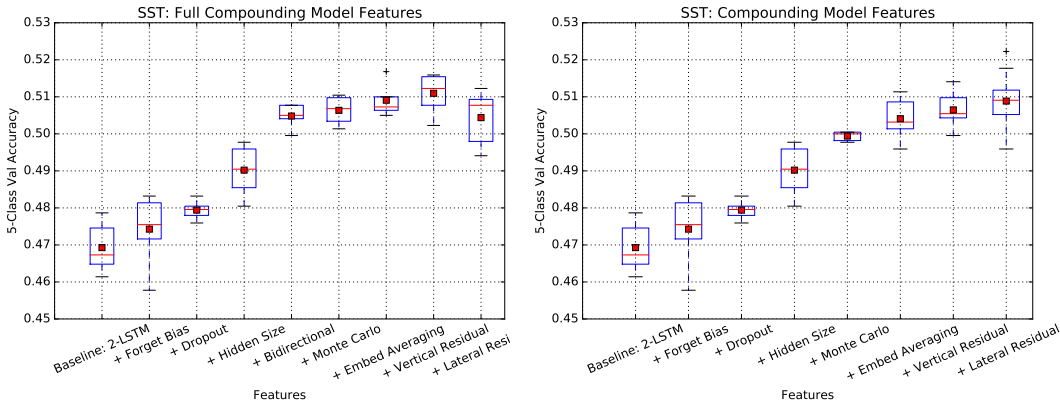

(a) Compounding feature models on 5-Class SST.

(b) Compounding feature models (minus bidirectional) for 5-Class SST.

Figure 4: These box-plots show the performance of compounding model features on fine-grain SST validation accuracy. The red points, red lines, blue boxes, whiskers and plus-shaped points indicate the mean, median, quartiles, range, and outliers, respectively.

We originally suspected that MC would provide marginal yet consistent improvements across datasets, while embed average pooling would especially excel for long sequences like in IMDB, where $n$-gram based models and deep unordered compositions have benefited from their ability to retain information from disparate parts of the text. The former hypothesis was largely confirmed. However, while embed average pooling was generally performance-enhancing, the performance boost it yielded for IMDB was not significantly larger than the one it yielded for SST, though that may have been because the other enhancements already encompassed most of the advantages provided by deep unordered compositions.

The only evident exceptions to the positive trend are the variations of residual connections. Which of Res-V1 (vertical only) and Res-V2 (vertical and residual) outperformed the other depended on the dataset and whether the network was bidirectional. The Res-V2 architecture dominated in experiments 4b and 5 while the Res-V1 (only vertical residuals) architecture is most performant in Figure 4a. This

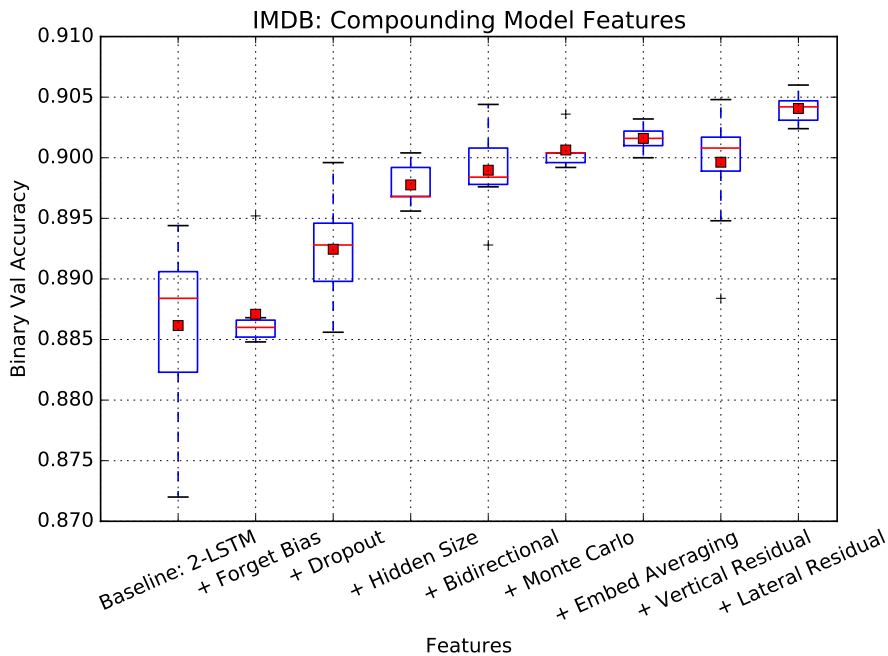

Figure 5: These box-plots show the performance of compounding model features on binary IMDB validation accuracy.

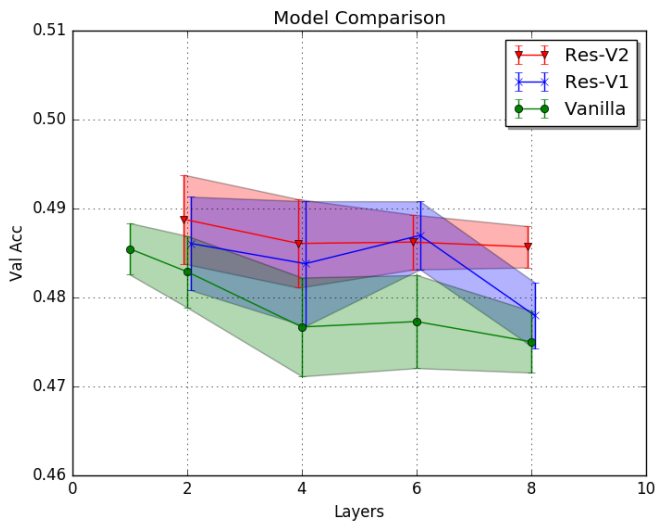

Figure 6: Comparing the effects of layer depth between Vanilla RNNs, Res-V1 and Res-V2 models on fine-grained sentiment classification (SST). As we increase the layers, we decrease the hidden size to maintain equivalent model sizes. The points indicate average validation accuracy, while the shaded regions indicate 90% confidence intervals.

suggests for short sequences, bidirectionality and lateral residuals conflict. Further analysis of the effect of residual connections and model depth can be found in Figure 6. In that figure, the number of parameters, and hence model size, are kept uniform by modifying the hidden size as the layer depth changed. The hidden sizes used for 1, 2, 4, 6, and 8 layer models were 250, 170, 120, 100, and 85 respectively, maintaining $\approx 550,000$ total parameters for all models. As the graph demonstrates,

| Model | # Params (M) | Train Time / Epoch (sec) | Test Acc (%) |
|---|---|---|---|
| RNTN (Socher et al., 2013) | – | – | 45.7 |
| CNN-MC (Kim, 2014) | – | – | 47.4 |
| DRNN (Irsoy and Cardie, 2014) | – | – | 49.8 |
| CT-LSTM (Tai et al., 2015) | 0.317 | – | 51.0 |
| DMN (Kumar et al., 2016) | – | – | **52.1** |
| NTI-SLSTM-LSTM (Munkhdalai and Yu, 2016) | – | – | **53.1** |
| Baseline 2-LSTM | 0.553 | $\approx 2,100$ | 46.4 |
| Large 2-LSTM | 8.650 | $\approx 3,150$ | 48.7 |
| Bi-2-LSTM | 8.650 | $\approx 6,100$ | 50.9 |
| Bi-2-LSTM+MC+Pooling+ResV | 8.740 | $\approx 8,050$ | **52.2** |
| 2-LSTM+MC+Pooling+ResV+ResL | 8.740 | $\approx 4,800$ | **51.6** |

Table 1: Test performance on the Stanford Sentiment Treebank (SST) sentiment classification task.

| Model | # Params (M) | Train Time / Epoch (sec) | Test Acc (%) |
|---|---|---|---|
| SVM-bi (Wang and Manning, 2012) | – | – | 89.2 |
| DAN-RAND (Iyyer et al., 2015) | – | – | 88.8 |
| DAN (Iyyer et al., 2015) | – | – | 89.4 |
| NBSVM-bi (Wang and Manning, 2012) | – | – | **91.2** |
| NBSVM-tri, RNN, Sentence-Vec Ensemble (Mesnil et al., 2014) | – | – | **92.6** |
| Baseline 2-LSTM | 0.318 | $\approx 1,800$ | 85.3 |
| Large 2-LSTM | 2.00 | $\approx 2,500$ | 87.6 |
| Bi-2-LSTM | 2.00 | $\approx 5,100$ | 88.9 |
| Bi-2-LSTM+MC+Pooling+ResV+ResL | 2.08 | $\approx 5,500$ | 90.1 |

Table 2: Test performance on the IMDB sentiment classification task.

normal LSTMs ("Vanilla") perform drastically worse as they become deeper and narrower, while Res-V1 and Res-V2 both see their performance stay much steadier or even briefly rise. While depth wound up being far from a panacea for the datasets we experimented on, the ability of an LSTM with residual connections to maintain its performance as it gets deeper holds promise for other domains where the extra expressive power provided by depth might prove more crucial.

Selecting the best results for each model, we see results competitive with state-of-the-art performance for both IMDB[1] and SST, even though many state-of-the-art models use either parse-tree information (Tai et al., 2015), multiple passes through the data (Kumar et al., 2016) or tremendous train and test-time computational and memory expenses (Le and Mikolov, 2014). To our knowledge, our models constitute the best performance of purely sequential, single-pass, and computationally feasible models, precisely the desired features of a solid out-of-the-box baseline. Furthermore, for SST, the compounding enhancement model without bidirectionality, the final model shown in Figure 4b, greatly exceeded the performance of the large bidirectional model (51.6% vs 50.9%), with significantly less training time (Table 1). This suggests our enhancements could provide a similarly reasonable and efficient alternative to shared-weight bidirectionality for other such datasets.

# 7 CONCLUSION

We explore several easy to implement enhancements to the basic LSTM network that positively impact performance. These include both fairly well established extensions (biasing the forget gate, dropout, increasing the model size, bidirectionality) and several more novel ones (Monte Carlo

---

[1]For IMDB, we benchmark only against results obtained from training exclusively on the labeled training set. Thus, we omit results from unsupervised models that leveraged the additional $50,000$ unlabeled examples, such as Miyato et al. (2016).

model averaging, embed average pooling, residual connections). We find that these enhancements improve the performance of the LSTM in classification tasks, both in conjunction or isolation, with an accuracy close to state of the art despite being more lightweight and using less information than the current state of the art models. Our results suggest that these extensions should be incorporated into LSTM baselines.

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
