# Peer review of "A Way out of the Odyssey: Analyzing and Combining Recent Insights for LSTMs"

_ICLR 2017 — rejected_

[Reviewer Comment · AnonReviewer3 · 11 Dec 2016]
**Datasets**

I find the experiments not convincing because the two datasets are quite similar (sentiment analysis). I was wondering if you have tried your proposed models/methods on much more different tasks (e.g. machine translation, question answering, etc.)

[Official Review · AnonReviewer1 · rating 5 · confidence 4 · 17 Dec 2016]

This paper presents three improvements to the standard LSTM architecture used in many neural NLP models: Monte Carlo averaging, embed average pooling, and residual connections. Each of the modifications is trivial to implement, so the paper is definitely of interest to any NLP researchers experimenting with deep learning. 

With that said, I am concerned about the experiments and their results. The residual connections do not seem to consistently help performance; on SST the vertical residuals help but the lateral residuals hurt, and on IMDB it is the opposite. More fundamentally, there need to be more tasks than just sentiment analysis here. I'm not quite sure why the paper's focus is on text classification, as any NLP task using an LSTM encoder could conceivably benefit from these modifications. It would be great to see a huge variety of tasks like QA, MT, etc., which would really make the paper much stronger. 

At this point, while the experiments that are included in the paper are very thorough and the analysis is interesting, there need to be more tasks to convince me that the modifications generalize, so I don't think the paper is ready for publication.

[Official Review · AnonReviewer2 · rating 5 · confidence 4 · 17 Dec 2016]
**No Title**

I agree with the other reviewer that the application areas are limited in the paper. I agree with the overall sentiment of the paper to evaluate effectiveness of some of the more recent techniques in this area, in conjunction with the recurrent networks. 

The paper advertises itself as a method (or a list of methods) of improving the recurrent baselines when performing experiments, however fails (or not shown) to generalize to other tasks. Effectiveness of these methods need to be shown across a wide variety of tasks if we intend to replace traditional baselines in general, rather than a specific subset of applications.

I like the desire to evaluate many of the recent techniques and having many replications of experiments towards this end (which is a strong point of the paper). However, whether there are synergies of some of the enhancements with sentiment analysis or not, we cannot see from these results. It would be interesting to see whether some of these results generalize across a wide variety of tasks.

[Official Review · AnonReviewer3 · rating 5 · confidence 4 · 19 Dec 2016]
**official review**

The paper proposes and analyses three methods applied to traditional LSTMs: Monte Carlo test-time model averaging, average pooling, and residual connections. It shows that those methods help to enhance traditional LSTMs on sentiment analysis. 

Although the paper is well written, the experiment section is definitely its dead point. Firstly, although it shows some improvements over traditional LSTMs, those results are not on par with the state of the art. Secondly, if the purpose is to take those extensions as strong baselines for further research, the experiments are not adequate: the both two datasets which were used are quite similar (though they have different statistics). I thus suggest to carry out more experiments on more diverse tasks, like those in "LSTM: A Search Space Odyssey"). 

Besides, those extensions are not really novel.

[Final Decision · Program Chairs · 06 Feb 2017]
**ICLR committee final decision**

The paper attempts to perform an interesting exploration (how to combine different tricks for LSTM training) but does not take it far enough. 
 
 Pros:
 - interesting attempt at studying different techniques to improve LSTM training results
 Cons:
 - not very strong baselines
 - limited set of domains were explored
 - low in novelty (which wouldn't be a problem if the comparison was more thorough -- see above 2 points).